# Reducing Heavy Metal Contamination in Soil and Water Using Phytoremediation

**DOI:** 10.3390/plants13111534

**Published:** 2024-06-01

**Authors:** Yryszhan Zhakypbek, Bekzhan D. Kossalbayev, Ayaz M. Belkozhayev, Toktar Murat, Serik Tursbekov, Elaman Abdalimov, Pavel Pashkovskiy, Vladimir Kreslavski, Vladimir Kuznetsov, Suleyman I. Allakhverdiev

**Affiliations:** 1Department of Mine Surveying and Geodesy, Institute Mining and Metallurgical Institute named after O.A. Baikonurov, Satbayev University, Almaty 050043, Kazakhstan; s.tursbekov@satbayev.university (S.T.); eabdalimov@gmail.com (E.A.); 2Ecology Research Institute, Khoja Akhmet Yassawi International Kazakh-Turkish University, Turkistan 161200, Kazakhstan; kossalbayev.bekzhan@gmail.com; 3Faculty of Biology and Biotechnology, Al-Farabi Kazakh National University, Al-Farabi Ave. 71, Almaty 050038, Kazakhstan; belkozhayev1991@gmail.com; 4M.A. Aitkhozhin Institute of Molecular Biology and Biochemistry, Almaty 050012, Kazakhstan; 5Department of Agronomy and Forestry, Faculty of Agrotechnology, Kozybayev University, Petropavlovsk 150000, Kazakhstan; murat-toktar@mail.ru; 6Department of Soil Ecology, Kazakh Research Institute of Soil Science and Agrochemistry named after U.U. Uspanov, Al-Farabi Ave. 75, Almaty 050060, Kazakhstan; 7K.A. Timiryazev Institute of Plant Physiology, Russian Academy of Sciences, Botanicheskaya Street 35, Moscow 127276, Russia; vlkuzn@mail.ru (V.K.); suleyman.allakhverdiev@gmail.com (S.I.A.); 8Institute of Basic Biological Problems, Russian Academy of Sciences, Pushchino 142290, Russia; vkreslav@rambler.ru

**Keywords:** contaminant detoxification, heavy metals, hyperaccumulators, phytoremediation, rhizosphere, transporters

## Abstract

The increase in industrialization has led to an exponential increase in heavy metal (HM) soil contamination, which poses a serious threat to public health and ecosystem stability. This review emphasizes the urgent need to develop innovative technologies for the environmental remediation of intensive anthropogenic pollution. Phytoremediation is a sustainable and cost-effective approach for the detoxification of contaminated soils using various plant species. This review discusses in detail the basic principles of phytoremediation and emphasizes its ecological advantages over other methods for cleaning contaminated areas and its technical viability. Much attention has been given to the selection of hyperaccumulator plants for phytoremediation that can grow on heavy metal-contaminated soils, and the biochemical mechanisms that allow these plants to isolate, detoxify, and accumulate heavy metals are discussed in detail. The novelty of our study lies in reviewing the mechanisms of plant–microorganism interactions that greatly enhance the efficiency of phytoremediation as well as in discussing genetic modifications that could revolutionize the cleanup of contaminated soils. Moreover, this manuscript discusses potential applications of phytoremediation beyond soil detoxification, including its role in bioenergy production and biodiversity restoration in degraded habitats. This review concludes by listing the serious problems that result from anthropogenic environmental pollution that future generations still need to overcome and suggests promising research directions in which the integration of nano- and biotechnology will play an important role in enhancing the effectiveness of phytoremediation. These contributions are critical for environmental scientists, policy makers, and practitioners seeking to utilize phytoremediation to maintain the ecological stability of the environment and its restoration.

## 1. Introduction

Environmental contaminants harmful to human health often stem from anthropogenic activities, contributing to pollution in air, water, and soil. Natural phenomena such as volcanic eruptions can also lead to environmental pollution. Humans are exposed to these pollutants through inhalation, ingestion, and dermal contact, with “dose” referring to the quantity of pollutant intake. This dose is influenced by the exposure duration and concentration, leading to varied health outcomes. While industrialization promotes development, it also significantly increases pollutant levels in the environment, posing widespread health risks globally [1].

The intensification of anthropogenic activities has led to irreversible environmental damage, resulting in a surge in soil contamination by heavy metals. This trend has sparked widespread concern within the scientific community. HMs, namely, cadmium (Cd), zinc (Zn), chromium (Cr), nickel (Ni), and lead (Pb), which are known for their high atomic numbers and densities, exhibit remarkable persistence in the soil without undergoing decomposition [2]. Typically, the prevalence of HMs under natural conditions remains low, necessitating prolonged periods of exposure to elevated concentrations. Although HMs can naturally occur in the environment, human activities such as oil extraction, chemical fertilizer application, wastewater discharge, pesticide use, ore exploitation, and fossil fuel combustion are the primary sources of HM pollution [3].

HMs are classified into essential and nonessential HMs based on their roles in the natural world. While essential HMs are vital in small quantities, high concentrations can be toxic or detrimental to human health. On the other hand, nonessential HMs, although less crucial to organisms, can cause severe toxicity if they accumulate in plant cells. The presence of both HMs in agricultural areas can negatively impact plant biochemistry and physiology, leading to reduced productivity and potential health risks for consumers [2].

To prevent HM ingress into the soil, the atmosphere, and aquatic ecosystems and to revitalize degraded soil, various techniques have been established. These techniques can be broadly classified into mechanical and chemical interventions, such as soil incineration, waste disposal, and the establishment of electric zones [4]. However, chemical methods have certain drawbacks, including high costs, limited contaminant control, residual chemical persistence, irreversible soil composition changes, soil cover modifications, and potential secondary contamination. Therefore, the urgent development of cost-efficient and ecologically sound methodologies for the efficient remediation of HM-contaminated areas is necessary [5].

Phytoremediation, an eco-friendly method utilizing plants to extract or reduce HM levels in contaminated soil, has gained prominence [6]. Notably, even in soils with minimal HMs, plants can absorb these elements through their roots, facilitating soil restoration. Plants establish complex underground rhizosphere ecosystems, facilitating HM absorption, promoting biological metabolism, rejuvenating the soil, and fostering beneficial microorganisms [7].

Recent research efforts have focused on elucidating the intricate mechanisms governing HM uptake by plants and developing effective phytoremediation technologies for soil revitalization. This review explored the intricate mechanisms underlying HM absorption by plants thriving in contaminated environments. In addition, this review examines strategies employed by plants to shield themselves from HM exposure, expedite detoxification, and enhance resilience under challenging conditions. Recent advancements in phytoremediation technologies have been highlighted, emphasizing the mechanisms governing HM absorption by plants and innovative approaches to enhancing their biological susceptibility and resistance to HMs. This comprehensive review also serves as a platform for addressing the prevailing challenges and outlines the promising prospects of phytoremediation as a vital tool in environmental restoration [4].

## 2. Phytoremediation Types

The selection of suitable methods for the successful implementation of phytoremediation is highly important. Within scientific research, four distinct phytoremediation methods are recognized: phytoextraction, rhizofiltration, phytostabilization, phytodegradation, and phytovolatilization. Phytofiltration encompasses techniques such as rhizofiltration and blastofiltration, aimed at removing contaminants from water using plant systems. Rhizofiltration specifically involves the use of plant roots to absorb metals and other pollutants directly from water. On the other hand, blastofiltration uses certain plant seedlings for the removal of heavy metals like lead and cadmium from water, leveraging the plant’s ability to absorb and decrease the concentration of these metals, thus purifying the water effectively. Both techniques offer ecological and efficient methods for treating contaminated water bodies [8]. And phytostimulation leverages the symbiotic relationship between plants and soil microorganisms, which can improve the bioavailability of heavy metals and stimulate plant growth. Techniques include the use of specific plant species, genetic engineering, and the application of growth-promoting rhizobacteria to increase the efficiency of phytoremediation processes [9]. The selection of an appropriate method tailored to specific goals can streamline research efforts and greatly enhance the achievement of desired results [5]. Phytoremediation, a dynamic field within environmental science, harnesses the innate capabilities of plants to mitigate contaminants in soil, water, and air. Proficient application of the four phytoremediation techniques is pivotal for optimizing the outcomes of any restoration project. When selecting the most suitable phytoremediation approach, it is crucial to consider factors such as contaminant type and concentration, local environmental conditions, and project objectives. Furthermore, ongoing research in phytoremediation continues to unveil new plant species and hybrid techniques, thereby expanding the arsenal available to environmental restoration experts [10] (Figure 1).

Table 1 provides an overview of the advantages and limitations of various phytoremediation methods, offering insights into their suitability for diverse scenarios and contaminants. These phytoremediation methods offer a sustainable and environmentally friendly alternative to conventional restoration techniques [6]. The choice of method depends on several factors, including the type and concentration of pollutants, site conditions, plant adaptability, and restoration objectives. Phytoextraction has been demonstrated to be a cost-effective and environmentally friendly method that is particularly effective for accessing metals such as Pb, Zn, and Cd, as highlighted by Bosiacki et al. (2014) and Corzo Remigio et al. (2020) [11,12]. Rhizofiltration, as discussed by Abdullahi 2015, offers a viable solution for water remediation, efficiently addressing contaminants such as Cr and Cd [13]. The potential of phytostabilization for reducing contaminant migration, detailed in studies by Bolan et al. (2011) and Limmer and Burken (2016), confirms its utility for metals such as Cu and As [14,15]. Finally, the application of phytovolatilization in eradicating organic pollutants, including chlorinated solvents according to Muthusaravanan et al. (2020), showcases the broad applicability of phytoremediation strategies [16]. One of the important examples of phytovolatilization is the application of transgenic plants for soil decontamination from organic mercury-containing compounds (methylmercury, dimethylmercury, and methylmercury chloride) [17]. This comprehensive analysis underscores the significance of selecting suitable phytoremediation techniques based on specific site conditions and contaminant types, paving the way for future advancements in environmental restoration efforts.

Future work should continue to explore genetic engineering to enhance plant capabilities, synergistic phytoremediation strategies, and the integration of microbial remediation to improve plant efficiency and sustainability.

### 2.1. Phytoextraction

Phytoextraction involves plants extracting contaminants, primarily HMs, from the soil. This technique has proven particularly effective in areas contaminated with elements such as Pb, Cd, and Zn. Certain plants, termed hyperaccumulators, possess the unique ability to hyperaccumulate HMs within their tissues. Hyperaccumulation is the process by which certain plants, known as hyperaccumulators, absorb and concentrate exceptionally high levels of heavy metals from the soil into their tissues. These plants can tolerate and sequester metals such as cadmium, nickel, lead, and zinc, often to levels 100 times greater than typical plants [9]. These plants are cultivated in contaminated soil where they absorb and accumulate HMs in aboveground biomass (AGB), such as leaves and stems. Once plants have accumulated substantial amounts of contaminants, they are harvested and safely disposed of, effectively reducing the concentration of contaminants in the soil [18].

Phytoextraction of heavy metals involves several phases comprising the mobilization of metal cations in the rhizosphere followed by their absorption and translocation from the roots to the aboveground shoots of the plant and, consequently, the deposition and compartmentalization of heavy metal ions inside the plant tissues [1]. When employing bacterial phytoextraction for the remediation of heavy metal-contaminated soil, there is a continuous occurrence of biochemical processes, encompassing metabolism, energy conversion, and information exchange among microbes, plant roots, and the rhizosphere environment. These processes significantly shape the metabolite composition within the rhizosphere, which is predominantly characterized by organic compounds of low molecular weight, such as carbohydrates, fatty acids, amino acids, lipids, and various other chemicals. Metabolites present in the rhizosphere undergo alterations due to the physiological and biochemical adaptations of plants and microbes to heavy metals, thus playing a crucial role in the conversion and movement of these metals within the plant–soil ecosystem [5]. Wang et al. (2020) critically reviewed field trials of phytomining and phytoextraction, highlighting the influencing factors and effects of additives on the enhancement of phytoextraction efficiency [19].

### 2.2. Rhizofiltration

Rhizofiltration, a phytoremediation method designed specifically for water purification, has demonstrated remarkable efficacy in eliminating HMs and organic compounds from contaminated water sources. Specifically, plants with a strong affinity for certain contaminants are cultivated hydroponically in water bodies contaminated with these substances. Plant roots actively absorb contaminants from water. This natural filtration process relies on the unique ability of plants to absorb and accumulate contaminants. Regular harvesting of plants is essential for maintaining the effectiveness of rhizofiltration. This continuous cycle of growth, absorption, and harvest ensures the ongoing purification of contaminated water [20]. Rhizofiltration is an eco-friendly and sustainable method for water treatment that leverages the natural detoxification capabilities of plants. It is particularly valuable in scenarios where conventional treatment approaches may prove impractical or excessively costly [21].

Rhizofiltration stands out for its cost-effectiveness compared to traditional water treatment approaches by leveraging the innate capabilities of plants to purify contaminated water, thus reducing the need for mechanical and chemical interventions. The primary expenses associated with rhizofiltration pertain to the initial setup of plant systems and their ongoing maintenance, which are generally lower than the operational and upkeep costs associated with mechanical treatments. Rhizofiltration is an environmentally friendly and economically viable alternative for the removal of heavy metals and pathogens from wastewater, as demonstrated by Odinga et al. (2019) in a study involving *Phragmites australis* and *Kyllinga nemoralis* [22]. Sikhosana et al. (2020) further validated the efficiency of rhizofiltration for nitrogen removal from urban runoff, emphasizing its adaptability and cost efficiency under various environmental conditions [23]. Wang et al. (2020) also discussed the application of rhizofiltration in treating water contaminated with heavy metals, emphasizing the potential of certain aquatic plants to clean up polluted water bodies [19]. Chirakkara et al. (2016) investigated the rhizofiltration potential of *Typha latifolia* in removing heavy metals from wastewater and demonstrated the effectiveness of the plant in water purification [24]. According to Benavides et al. (2018), *Zea mays* L. exhibited reductions of 13% in Hg, 28% in Pb, and 28% in Cr [25]. High rhizofiltration potential is demonstrated by the aquatic plant *Typha angustifolia*, which is capable of absorbing Cd and Zn at rates of 12 mg per plant and 58 mg per plant, respectively. With *Typha angustifolia* displaying a bioconcentration factor (BCF) value exceeding 100 and a lower translocation factor (TF) value, it has emerged as an excellent candidate for phytoremediation [21].

### 2.3. Phytostabilization

Phytostabilization serves as an ecologically benign method for impeding the dispersion of contaminants in soil, particularly in regions harboring metals or metalloids. Unlike the aim of extracting contaminants, the primary objective of phytostabilization is to immobilize contaminants within the soil. Specific plant species are carefully chosen for their ability to form bonds with soil contaminants, effectively restricting their movement to other areas or water bodies [1]. These plants establish a protective barrier that confines contaminants to their current location. Contaminated areas become less prone to the spread of harmful substances over time, rendering phytostabilization a valuable approach for mitigating soil contamination without resorting to costly excavation and removal approaches. In 2019, Monaci et al. explored the phytostabilization potential of *Erica australis* L. and *Nerium oleander* L. in the Riotinto mining area [22]. Another study by Fresno et al. (2018) evaluated an aided phytostabilization strategy that employs several composite amendments of iron sulfate and organic materials in combination with *Lupinus albus* L. for the remediation of soil contaminated with arsenic and copper [26]. Research has shown that the application of these amendments, especially when combined with the growth of white lupin, can effectively reduce the bioavailability of arsenic and copper in contaminated soils, suggesting a viable approach for the phytostabilization of contaminated lands [27].

Several techniques are used in phytostabilization to lessen the impact and dispersion of pollutants in soil. One important process is the absorption and retention of pollutants in plant root systems. Through their roots, plants draw toxins from the soil as they develop, removing these dangerous materials from the surrounding environment. Furthermore, phytostabilization alters soil properties that influence pollutant speciation and immobilization. To reduce the mobility and availability of pollutants, conditions that promote the binding or precipitation of these contaminants are created by manipulating factors such as pH, organic matter content, and redox levels [8]. Plant exudates from plant roots are also essential for phytostabilization because they include substances that interact with pollutants in the soil to help precipitate or immobilize them. Root exudates help control chemical reactions in the rhizosphere [28].

### 2.4. Phytovolatilization

Phytovolatilization is a phytoremediation process in which plants absorb contaminants through their roots and convert them into a gaseous form. These contaminants are then released into the atmosphere through the plant transpiration. This method is particularly useful for dealing with certain organic compounds and volatile heavy metals [1]. This process involves selecting plant species known for their ability to absorb volatile contaminants through the roots. Upon absorption, plants convert contaminants into less toxic or nontoxic gases within their tissues, but this can only apply to heavy metals, such as Hg and Se, which may exist in the environment as gaseous species, but this does not occur necessarily. These gases are subsequently released into the air through mechanisms such as transpiration, effectively decreasing their presence in the soil and preventing potential environmental harm. Phytovolatilization offers an eco-friendly approach to managing volatile contaminants, contributing to the overall remediation of contaminated sites and minimizing associated risks. Doty et al. (2007) demonstrated the effectiveness of genetically modified poplars for the phytovolatilization of trichloroethylene (TCE), a common groundwater contaminant [29].

Phytovolatilization methods are used to help remove volatile organic pollutants from soil and release them into the atmosphere. One such process that increases the availability of pollutants for absorption by plant roots is lowering the water table. Variations in the diel water table cause gas fluxes that facilitate the transport of volatile organic pollutants toward the soil surface, where plant tissues may absorb them [21]. Increased soil permeability promotes absorption by plant roots by facilitating the transport of volatile organic compounds through the soil matrix. Volatile organic pollutants are more easily accessed by plant roots through the soil profile due to chemical transfer via hydraulic redistribution. Furthermore, volatile organic pollutants are transported closer to plant roots by water advection toward the surface, which facilitates their absorption and subsequent volatilization [15].

## 3. Metal Absorption and Tolerance Mechanisms

Plants that grow in metal-contaminated environments have adapted to tolerate high concentrations of metals, primarily by limiting their transport to above-ground parts. However, specific species, termed hyperaccumulators, are notable for their capacity to concentrate metals in their shoots at concentrations significantly greater than those found in the surrounding environment. These hyperaccumulators are particularly valuable for the phytoremediation of soils contaminated with heavy metals due to human activities. They are also crucial for phytomining, the process of extracting economically valuable metals from metal-rich soils [1]. These species are characterized by their efficient uptake of heavy metals, their transport to the shoots, and their sequestration in forms that are benign to essential biochemical processes, notably photosynthesis. To combat heavy metal toxicity, plants have evolved sophisticated and specialized mechanisms, including the modification of rhizosphere interactions, the use of heavy metal transporters (HMTs), chelation processes, mechanisms for detoxification and tolerance, and strategies for the translocation, accumulation, and mitigation of reactive oxygen species (ROS) induced by metal stress [30].

### 3.1. Rhizosphere Interactions

In the rhizosphere, microbial communities play a crucial role in the phytoremediation process. Liu et al. (2020) showed that the presence of *Trifolium repens* L. in Cd-contaminated soil significantly influences the microbial community structure and metabolic activities, enhancing remediation efficiency [31]. This suggests that synergistic interactions between plants and microorganisms are vital for stabilizing plant–microbial ecosystems in contaminated environments.

The rhizosphere, where plant roots and soil interconnect, serves as a critical interface for understanding the intricate mechanisms governing plant metal uptake. Within this domain, complex interactions of chemical, physical, and biological processes regulate HM availability in the soil, impacting metal assimilation by plants [3]. A pivotal facet of rhizosphere dynamics is root exudation, which includes a diverse array of substances such as organic acids, sugars, amino acids, phenols, and enzymes. These exudates are crucial for modifying the soil environment, potentially increasing or decreasing the availability of metals to plants, thus influencing nutrient uptake and contaminant mobility. Manipulating these exudates enhances HM solubility, thus facilitating plant uptake. The exudate composition is contingent on the plant species and soil conditions [32]. Microorganisms in the rhizosphere engage in complex interactions with plant roots and root exudates, influencing HM mobility. Certain microorganisms catalyze HM solubilization, while others immobilize HMs through adsorption or precipitation, profoundly affecting metal bioavailability and uptake. Understanding rhizosphere interactions is integral to phytoremediation, as plant effectiveness in remediating HM-contaminated soils hinges on these processes. Scientists have undertaken efforts to manipulate and fine-tune rhizosphere complexities to enhance HM uptake by plants. In 2021, Mahmud et al. explored leveraging rhizosphere microorganisms to optimize phytoremediation outcomes [33].

The rhizosphere, an ever-evolving interface where plant roots, which have distinctive exudative traits and a diverse composition of soil microorganisms, synergistically impact HM availability, forms the foundation for plant efficacy as an eco-friendly phytoremediation tool. These intricate rhizosphere interactions offer sustainable solutions for alleviating the threats of environmental HM pollution [19].

When organisms are compatible, they may communicate and interact, which makes it easier for them to trade resources. This leads to the growth of a symbiotic relationship. The genetic, epigenetic, and biochemical molecular pathways that underlie plant–microbe interactions are essential for forming advantageous relationships that promote plant growth and stress tolerance. According to Ahkami et al. (2017), the molecular analysis of these interactions offers important insights into the genome, protein, and metabolite levels [34]. This allows for the manipulation of plants and microbes for the purpose of rhizosphere engineering, which in turn increases plant production.

### 3.2. Heavy Metal Transporters

Higher plants use complex systems that rely on a small number of transport pathways to take up metal ions and inorganic nutrients from the soil. Cotransport processes allow heavy metals to cross the plasma membrane of root cells. Since the biological activities of hazardous heavy metals such as Pb and Cd are unknown, they are also transported by these broad processes in the absence of specialist transporters. Heavy metal transporters (HMTs) are specialized proteins that play a critical role in the movement of metal ions within plant cells, facilitating their transfer through the xylem and among various cellular compartments. HMTs mainly assist in the efflux of heavy metal ions from the cytoplasm into organelles or across the plasma membrane in plants and several other hyperaccumulator species. While *Arabidopsis thaliana* is not a hyperaccumulator, other plants within the Brassicaceae family, such as *Noccaea caerulescens* (formerly known as *Thlaspi caerulescens*) and Alyssum species, are known for their hyperaccumulating properties. These plants can absorb and accumulate extremely high levels of heavy metals like zinc, nickel, and cadmium, making them valuable for phytoremediation efforts [35].

The intracellular milieu is saturated with potential ligands for the binding of heavy metal ions, which results in the buffering of intracellular concentrations of heavy metals to quite low levels, down to femto- or attomolar levels for Cu and Zn [36,37]. Additionally, the accumulation of a negative charge on the inner side of the plasmalemma facilitates the uptake of cations [38]. As a result, a steep concentration gradient of HM ions directed from the apoplast to the cytosol and from the vacuolar lumen to the cytosol generally occurs. Therefore, the uptake of HM ions from the extracellular milieu into the cytosol often occurs passively through transporters that facilitate diffusion processes, whereas HM efflux from the cytosol into either the apoplast or vacuolar lumen requires active transport mechanisms [39].

Among the transporters responsible for the passive uptake of HM ions, members of the Zn-regulated, iron-regulated transporter-like protein/iron-regulated transporter (ZIP/IRT) family are probably the best characterized. These membrane proteins have wide substrate specificity, transporting not only Zn^2+^ but also other essential metals (Fe^2+^, Mn^2+^, Co^2+^), as well as toxic Cd^2+^ ions [40]. This is a clear advantage in terms of phytoremediation, as is the fact that the main mechanism of the regulation of HM uptake by ZIP/IRT transporters is the regulation of the transcription of corresponding genes [41], thus making it easier to manipulate HM uptake by ZIP/IRT transporters using molecular genetics methods. The ions Fe^2+^ and Mn^2+^, which are less tightly bound by the intracellular milieu than Cu^2+^ and Zn^2+^, are transported into the cytosol in symport with protons by natural resistance-associated macrophage protein (NRAMP, such as NRAMP1) transporters [42]. Members of the NRAMP family located in tonoplasts and vesicles are involved in the remobilization of HM ions stored in these compartments back to the cytosol [43]. In addition to Fe^2+^ and Mn^2+^, NRAMP proteins can also transport Ni^2+^, Co^2+^, Cd^2+^, Pb^2+^, and other metal ions [42], making NRAMP proteins potential candidates for improving plant uptake of several metals simultaneously [44]. Some toxic HM ions are transported by proteins that are not normally involved in transition metal transport; for example, Pb can be transported through Ca-permeable cyclic nucleotide-gated channels (CNGCs), Cr(VI) by the sulfate transporter AtSultr1;2, and As can be taken up by phosphate and silicon transporters [45]. Many transporters that mediate HM uptake in the cell have quite limited specificity, which is probably compensated for by the tight regulation of transporters at the transcriptional level [41]. The copper is likely reduced by Cu^2+^-reducing FRO from Cu^2+^ to Cu^+^ outside of the cell and then transported through the plasmalemma by copper transporters (COPTs), which are highly specific to monovalent Cu^+^ ions, after which Cu+ is specifically delivered to intracellular targets by Cu chaperones [46], leaving free Cu^+^ cytosolic concentrations at very low (likely attomolar) levels [47]. Copper uptake is likely regulated more tightly than that of other essential transition metals, such as Zn, Mn, or Fe, due to the greater ability of Cu to bind to intracellular sites and the deleterious effects of such binding on intracellular processes. In addition to being directly taken up as Cu^+^ ions, Cu (and Mn and Fe) can be transported to the cytosol from both the extracellular medium and organelles in complex with nicotianamine and phytosiderophores by yellow stripe-like (YSL) proteins [48,49,50].

In contrast to HM uptake on the plasmalemma, HM efflux from the cell and HM transport from the cytosol to other cellular compartments must overcome both low intracellular levels of free HM ions and positive membrane potential outside of the cell; therefore, these processes must rely on energy-dependent transport. Cation diffusion facilitator/metal tolerance protein (CDF/MTP) transporters utilize proton gradients for the transport of metal ions, but in contrast to natural resistance-associated macrophage protein (NRAMP) transporters, CDF/MTPs are proton antiporters that efflux Zn^2+^, Fe^2+^, Co^2+^, Ni^2+^, Cd^2+^, and Mn^2+^ ions from the cytosol to either the apoplast or cellular compartments [48]. Instead of relying on the proton gradient, heavy metal ATPase (HMA) transporters directly utilize ATP energy to support the transport of HM ions out of the cytosol. There are two main subgroups of HMAs, namely, Pb/Zn/Cd/Co-transporting P_1B_-ATPases and Ag/Cu-transporter P_1B_-ATPases [51]. Interestingly, CDF/MTP transporters and HMA transporters can function in concert to mediate long-distance HM transport, as was demonstrated for Zn transport from metal-absorbing root cells to shoots [52]. Highly toxic Cd, As, and apparently Hg ions are sequestered in vacuoles in complex with phytochelatins (PCs), which are oligomers of glutathione produced by the enzyme phytochelatin synthase (PCS) (see below). The transport of these complexes from the cytosol to the vacuolar lumen is mediated by ATP-binding cassette (ABC) proteins [53,54].

### 3.3. Intracellular Ligands of Heavy Metals

After being acquired by the cell, HM ions are bound by different ligands, which is necessary not only to prevent their deleterious interactions with cellular macromolecules but also (for redox-active metals such as Cu, Fe, and Mn) to prevent them from participating in redox reactions via the formation of reactive oxygen species. Generally, various organic molecules can bind HM ions to a certain extent [55]. However, the actual importance of a given molecule in HM sequestration depends on both the stability of the complex with HM ions and the competitive effect of H^+^ binding to the molecule at a pH characteristic of the specific cellular compartment.

The N-containing compounds histidine and nicotianamine are likely the most universal HM chelators at pH levels close to neutral. Nicotianamine forms hexadendate and highly stable complexes with Fe, Cu, Ni, Co, Zn, and Mn at neutral or slightly acidic pH levels [41]. Nicotianamine participates in the cytosolic sequestration of HM ions, radial transport (in the root symplast), and long-distance axial transport via the phloem and xylem [56]. Histidine forms tridentate complexes with metal ions that are less stable than nicotianamine complexes, but due to the high histidine concentrations and relatively low metabolic cost of hisitidine compared to nicotianamine, this amino acid can play a major role in HM binding in plant cells, and both histidine and nicotianamine are utilized by hyperaccumulators to sequester and transport HM ions [56].

Cysteine-rich compounds, namely, glutathione and phytochelatins, can bind “soft” HMs with exceptionally high affinity. Phytochelatins effectively detoxify Cd by binding to Cd^2+^ ions in the cytosol with subsequent vacuolar transport of Cd-PC complexes by ABC transporters, where Cd-PCs are combined with sulfide and sequestered [57]. These complexes are likely able to dissociate, releasing Cd^2+^ for subsequent binding by organic acids in the vacuolar lumen [54]. Glutathione binds metals with lower affinity than phytochelatins and is likely involved in the binding of not only Cd but also Cu and Zn [56]. However, in contrast to nitrogen-containing compounds, sulfur-containing compounds play a minor role in metal sequestration in hyperaccumulators, probably due to the high cost of biosynthesis of these sulfur-containing compounds [56]. In addition to glutathione and phytochelatins, small metallothioneins also bind HM ions (primarily Zn, Cu, and Cd), but their role extends beyond metal detoxification since metallothioneins can function as metal chaperones, providing ions of essential metals to metal-dependent enzymes [57].

Organic acids represent the final major class of heavy metal-binding compounds in the cell. They bind metal ions with much lower affinities than nitrogen- or sulfur-containing compounds; however, due to the low metabolic cost of biosynthesis, high (millimolar) total concentrations, and the ability to bind metal ions even in acidic environments, organic acids, mainly citric and malic, are the major chelators of HM ions in the vacuolar lumen. Organic acids are utilized to bind heavy metals both for vacuolar storage and for long-distance transport in non-hyperaccumulation species and in hyperaccumulating plants [56].

In conclusion, the use of molecular genetics to improve the ability of plants to accumulate heavy metals faces several obstacles. Although many HM transporters, such as NRAMP or CDF/MTP, have wide substrate specificity, there is no single transporter that is effective in the transport of both “soft” (such as Cu, Hg, and As) and “hard” (such as Zn, Ni, Mn, and partly Cd) metals. Given that environmental contamination is generally polymetallic, no single genetically engineered line would be effective for the removal of all contaminating metals. Therefore, the accumulation of heavy metals can be detrimental to plants, and we need to simultaneously increase the ability to sequester and detoxify absorbed ions by increasing the accumulation of heavy metal chelators. Moreover, the accumulation of heavy metals is often restricted to roots in non-accumulating species, although this restriction can be overcome by activating the systemic root-to-shoot transport of heavy metals, as was demonstrated for the HMA4 transporter, which greatly increased root-to-shoot Zn transport in *A. thaliana* [52].

### 3.4. Phytochelatins

Phytochelatins (PCs) act as key gatekeepers, overseeing HM detoxification and regulation. These internal processes involve the formation of stable HM complexes within plant cells, effectively preventing the detrimental effects of HM toxicity while promoting HM sequestration and subsequent removal. A comprehensive understanding of chelation is imperative for developing effective phytoremediation strategies and improving plant tolerance to HM-induced stress [30].

Plants employ various chelating agents as their primary defense against the deleterious effects of HM toxicity. Notably, PC and metallothioneins play pivotal roles in biochemical defense. Phytochelatins demonstrate a remarkable affinity for HMs, controlling their binding and forming strong and stable complexes under precise enzymatic control. Moreover, metallothioneins, which are cysteine-rich, bind and sequester HM ions, further enhancing plant defense against toxicity [58].

Plant heavy metal detoxification and tolerance rely heavily on the chelation of metals inside the cytosol by high-affinity ligands. Amino acids, organic acids, and two peptide classes—PCs and metallothioneins—are examples of possible ligands. Among these peptides, a class of peptides called PCs, which have a generic structure called (γ-Glu-Cys)n-Gly, have been examined in great detail, especially in relation to their tolerance to cadmium. Exposure to heavy metals causes plants to quickly develop PCs, which are then produced without translation by PC synthase, an enzyme that is triggered by the presence of metal ions, using glutathione as a substrate. In addition, PC synthase genes have been identified in *A. thaliana* and yeast [59].

The synthesis of chelating agents involves intricate biochemical regulation. For instance, Cobbett 2000 conducted an extensive molecular investigation into a crucial HM detoxification process in *A. thaliana* [60]. This research sheds light on the intricate biochemical processes governing the formation of metal chelate complexes and emphasizes the importance of the detoxification process in augmenting plant-based metal mitigation strategies. Seregin and Kozhevnikova’s 2023 work elucidated the role of PCs, peptides rich in cysteine, in metal(loid) detoxification in plant cells, primarily by targeting cadmium and arsenic(III) [54]. Their work discusses PC biosynthesis, structure, and function in transporting and sequestering metal(loid)s within plant vacuoles, which are essential for understanding plant metal(loid) tolerance mechanisms.

The wave-like effects of chelation resonate deeply within plant cells, significantly influencing HM mobility. Chelating agents suppress HM toxicity by the formation of stable complexes, preventing HM intrusion into vital cellular processes. Additionally, these complexes act as overseers, managing HM movement and sequestration in plant tissues. This ensures reliable HM retention in vacuoles and other stored cell components [61]. A comprehensive understanding of the biochemical complexities and their far-reaching influence on HM mobility is needed to harness plant phytoremediation potential, thereby enhancing the mitigation of the pervasive threat of environmental HM contamination.

### 3.5. Detoxification and Tolerance Mechanisms

Considering plant life, the persistent challenges posed by HM-induced stress catalyze the development of detoxification and tolerance mechanisms. These strategies involve diverse biochemical and physiological processes calibrated to mitigate the deleterious impacts of HM exposure while maintaining vital cellular functions [31,32]. Enzymatic detoxification is the foremost among these mechanisms, wherein plants employ a range of enzymes as frontline defenses against invading HMs. Examples of these enzymes include glutathione S-transferases, which help in conjugating toxic metals to glutathione, facilitating their sequestration; superoxide dismutases, which mitigate oxidative stress caused by metals; and phytochelatin synthases, which synthesize phytochelatins that bind metals and enhance their vacuolar storage [62]. Metallothioneins and glutathione-S-transferases are crucial in the detoxification of heavy metals within plants. Metallothioneins bind to heavy metals, reducing their reactivity and toxicity. Glutathione-S-transferases, on the other hand, facilitate the conjugation of toxic metals with glutathione, aiding in their sequestration and removal from sensitive cellular areas [63].

Exposure to HMs causes a plant response, adapting to adverse conditions. This response involves the activation of stress response pathways characterized by the induction of stress-related genes and the synthesis of stress proteins. These complex reactions help plants repair HM-induced damage and ensure plant survival under metal stress. Morkunas et al. (2018) explored HM-induced signaling pathways and stress-responsive gene regulation in the plant kingdom, offering invaluable insights into how plants perceive and organize responses to severe HM-induced stress at the molecular level [64]. Understanding these responses is pivotal in strategic endeavors to advance phytoremediation initiatives and enhance tolerance in crops cultivated in contaminated soil. Enzymatic detoxification, compartmentalization, and the dynamic activation of stress response pathways represent adaptive traits and fertile ground for the development of eco-friendly strategies against HM contamination [65].

### 3.6. Translocation and Accumulation

This process begins with the uptake of heavy metals into plant roots and is driven by specialized transporters that facilitate the entry of metal ions into root cells. Once in the roots, metals can move along different trajectories. The HM pathway within plants, from absorption through roots to targeted accumulation in specific plant tissues, is a multifaceted phenomenon [58]. Understanding the mechanisms governing HM migration and accumulation is crucial for optimizing the potential of plants for phytoremediation to combat HM contamination. This process commences with HM uptake into plant roots, wherein some HMs traverse the xylem directly into the AGB, while others undergo temporary retention in root tissues or form complexes with chelating agents for subsequent translocation into the AGB [66].

HM accumulation in specific plant tissues occurs selectively and depends on the plant species. Certain plants preferentially accumulate HMs in their roots, acting as “scavengers” to prevent their upward movement to AGB. Conversely, “hyperaccumulators” demonstrate an exceptional ability to accumulate extremely high HM concentrations, often within their leaves or stems.

HM uptake and accumulation in plants are influenced by several factors, including HM chemical properties, plant species, soil conditions, and the presence of competing ions. Papoyan and Kochian (2004) pioneered the elucidation of the intricacies of HM absorption across different plant species, shedding light on the pivotal role of HMTs and revealing diverse patterns of HM accumulation in plant tissues, thereby providing invaluable insights for enhancing the phytoremediation process [67]. Furthermore, explorations of hyperaccumulator species such as *N. caerulescens* (earlier known as *T. caerulescens)* have revealed the genetic and physiological foundations of their exceptional ability to accumulate HMs. HM accumulation in plants is intricately regulated and complex, suggesting that plant phytoremediation has the potential for remediating HM-contaminated environments [68].

### 3.7. Physiological and Biochemical Aspects of Phytoremediation

Recent research has provided significant insights into these mechanisms, highlighting the potential of phytoremediation to not only cleanse the environment of pollutants but also contribute to ecological restoration and biodiversity conservation. The mechanisms underlying phytoremediation are diverse, and efforts to enhance the process are ongoing. Similarly, Rostami and Azhdarpoor (2019) discussed how plant growth regulators, such as auxins and gibberellins, can improve phytoremediation efficacy by increasing plant biomass and mitigating the adverse effects of contaminants [69]. To address the limitations of phytoremediation, including the slow growth of hyperaccumulator plants and low heavy metal activity in the soil, Shuang et al. (2021) explored strengthening technologies such as chemical and microbial enhancements [70]. These approaches aim to improve the efficiency of phytoremediation, making it a more viable and effective solution for environmental decontamination.

## 4. Plant Selection

One of the conditions for efficient phytoremediation of HM-contaminated areas is biological availability for plants, i.e., phytoavailability. When HMs are closely bound to soil organic matter, they might remain nonbioavailable to plants for a long period. Moreover, water-soluble HMs might traverse the root system without significant accumulation, emphasizing the importance of selecting plant species with anatomical, structural, physiological, and biochemical adaptations that confer resistance to adverse environmental conditions. The numerous plant species that thrive in contaminated areas may accumulate HMs but might not withstand unfavorable environmental conditions such as high temperatures, limited rainfall, and salinization. Thus, when evaluating phytoremediation methods, it is crucial to consider not only the moisture retention capability of plants but also their ability to adapt to local climatic conditions [71].

Plant selection is a crucial factor for ensuring successful phytoremediation. Native plant species are often preferred for phytoremediation because they require minimal maintenance and are well-adapted to the local climate. Metallophytes are plants, which are well adapted to HM soils and able to survive in heavy metal-rich soils. Metallophytes can be divided into some categories: (a) metal excluders that accumulate HMs accumulating mainly in roots; (b) metal indicators that accumulate HMs in their aerial parts; (c) metal accumulators accumulating high HM concentrations mainly in the aboveground plant parts, such as shoots and leaves [72]. Examples include *Alyssum bertolonii*, which hyperaccumulates nickel, and *N. caerulescens*, which is known for absorbing high levels of zinc and cadmium. *Armeria maritima* also thrives in environments rich in copper and lead. While excluders like certain populations of Silene vulgaris limit metal uptake to avoid toxicity, they often grow in metal-rich soils but are not necessarily metallophytes. These plants are valuable for their potential in phytoremediation, the process of using plants to clean up soil and water contaminated with metals. On the other hand, hyperaccumulators are capable of thriving in soils with high HM concentrations and accumulating HMs. Some hyperaccumulators specialize in the accumulation of specific HMs, while others can accumulate multiple HMs. Notably, hyperaccumulation is predominantly associated with Ni, which accounts for approximately 75% of known hyperaccumulator plants [73].

Currently, soil restoration in anthropogenically damaged areas, encompassing desert and semidesert lands, is an increasingly relevant global concern in the realm of phytoremediation. Hence, the use of halophytic hyperaccumulator plant varieties resistant to adverse environmental conditions is highly effective for this purpose. More than 70% of the industrial sites in Kazakhstan are situated in desert and semidesert regions, where xerophytic and halophytic plants are widely distributed. Therefore, identifying and examining hyperaccumulators among xerophytic and halophytic plant species is highly important [74].

The phytoremediation potential of halophytes is significant due to their ability to thrive in saline environments and economically manage metal contamination. These plants are effective at accumulating substantial amounts of heavy metals, offering a sustainable solution for improving soil quality in contaminated areas [75]. In addition, halophytes can be cultivated in arid and salt-affected regions [76].

Incorporating examples of hyperaccumulator halophytic and xerophytic plants can indeed enhance our understanding of phytoremediation. For instance, *Salsola kali* is a halophyte known for accumulating heavy metals, while *Atriplex canescens*, a xerophytic species, also shows capabilities for heavy metal uptake in contaminated soils. These species not only tolerate extreme conditions but also assist in the recovery of degraded environments through the accumulation of contaminants [76,77].

Desert-adapted plants with HM bioaccumulation capabilities are the most suitable candidates for phytoremediation in these areas. The bioindication and phytoremediation potential of lemongrass, Siam weed, wild grasses, vetiver, *Sesbania* spp., *Avena*, *Crotalaria*, and *Calotropis procera* have been extensively studied in desert regions. Indeed, the bioaccumulating potential of *C. procera* has been confirmed. Leveraging metallo-halophytes from Central Asia for soil reclamation in dry climates represents a practical and cost-effective strategy. The role of hyperaccumulator plants, particularly halophytic and xerophytic species, is crucial in protecting soil ecological functions. These plants are recognized for their ability to combat global issues like desertification and soil contamination by accumulating heavy metals, thereby contributing to ecological restoration and soil stability. This makes them valuable indicators of environmental health and sustainability [78].

Hyperaccumulator plants in desert and semidesert zones fulfill various functions beyond soil decontamination, including absorbing HMs into roots, stems, and leaves; improving soil fertility; enhancing food and human/animal safety; promoting environmental biodiversity; reducing carbon dioxide and dust emissions; creating a microclimate under arid conditions; and mitigating global warming (Figure 2). Hyperaccumulator plants are primarily recognized for their ability to remove heavy metals from soils and not for controlling phytopathogenic bacteria. However, certain beneficial interactions between hyperaccumulators and soil microorganisms might indirectly affect soil pathogen populations. Hyperaccumulator legume plants capable of nodule formation in desert and semidesert ecosystems are not very common. An example is Astragals, a genus known for some species that can accumulate selenium and are found in arid regions. These plants use their nodulation capabilities to fix nitrogen while also managing to grow in metal-rich soils, contributing to soil health in challenging environments [79,80].

Anthropogenic activities, such as industrial processes, mining, and agriculture, contribute to the accumulation of Pb, Cd, Hg, and As in the soil, necessitating effective remediation approaches. In recent years, phytoremediation has emerged as an eco-friendly and cost-effective method. Notably, vetiver grass, whose deep root system reaches up to 3 m, has displayed remarkable potential for stabilizing soil and absorbing contaminants, making it an ideal candidate for phytoremediation. Wei et al. (2011) demonstrated the efficacy of vetiver grass on HM accumulation from Pb and Zn mine waste, highlighting the significant HM sequestration in the roots [71]. This grass is increasingly employed in various projects to rehabilitate contaminated soil, enhance agricultural yield, and safeguard groundwater resources. Zeng et al. (2019) explored the phytoextraction potential of *P. vittata* when *P. vittata* was planted alongside trees such as *Morus alba* or *Broussonetia papyrifera* in As-, Cd-, Pb-, and Zn-contaminated soil [82]. These findings revealed that intercropping not only mitigates the toxic effects of metal(loid)s on plant growth but also enhances overall phytoextraction, suggesting a promising strategy for rehabilitating metal(loid)-contaminated soil. Furthermore, the profound root system of vetiver grass creates a conducive environment for the proliferation of arbuscular mycorrhizal fungi, which are vital components of the phytoremediation process. In symbiosis with plant roots, these fungi enhance nutrient absorption and plant growth and enhance resistance to contaminants and environmental stressors. Additionally, these plants produce biomass and augment the activities of beneficial soil microorganisms, further increasing phytoremediation efficiency. Although the effectiveness of arbuscular mycorrhizal fungi in phytoremediation may vary depending on the specific contaminants, plant species involved, and local environmental conditions, their overall contribution to plant productivity remains evident [83] (Figure 3).

## 5. Hyperaccumulating Plants: Mechanisms of Hyperaccumulation

The term “hyperaccumulator” describes a number of plants that are able to grow on metalliferous soils and concentrate elevated amounts of heavy metals in aerial organs [35]. The aerial organs of nearly 800 plant species can accumulate metals to levels that exceed the normal levels of other plants grown in the same habitat hundreds or thousands of times [85,86]. For example, hyperaccumulators of Ni can accumulate elevated amounts of Ni (>1000 mg kg^−1^ dry weight) in their shoots, and symptoms of toxicity are absent, whereas normal plants accumulate 0.01–5 mg kg^−1^ dry weight [87]. In addition, plants can effectively detoxify these metals, transforming them into less harmful forms. In some cases, detoxification can be effective in the root system. Thus, high resistance to Ni in plant populations of the hyperaccumulator *N. caerulescens* from serpentine soils was found, which is explained by the high efficiency of Ni detoxification in the roots and is not directly related to the efficiency of its translocation from roots to shoots [73]. Plants that are hyperaccumulators of heavy metals are good models for studying the mechanisms regulating heavy metal accumulation and adaptation under various environmental conditions. Two key hyperaccumulator plants from the cabbage family, *Arabidopsis halleri* and *N. caerulescens*, have been used in many works to study the mechanisms by which plants can accumulate significant amounts of HMs and survive [73,86,88]. However, worldwide challenges in identifying hyperaccumulator plants exist, resulting in the lack of a unified plant selection system and a lack of information on their distribution within various regions.

The key processes involved in the hyperaccumulation of heavy metals by hyperaccumulator plants in soil include the following [89]:

(a) Bioactivation of HMs in the rhizosphere by root–microbe interactions, which transform HMs into more available forms for absorption by the root system; (b) enhanced uptake by metal transporters in the plasma membrane; (c) detoxification of metals by distribution to the apoplast area, where they are linked to cell walls, and chelation of heavy metals in the cytoplasm with various ligands, such as phytochelatins, metallothioneins, and metal-binding proteins; (d) sequestration of metals into the vacuole by tonoplast-located transporters. The increasing application of different molecular genetic technologies has led to a better understanding of the mechanisms underlying the accumulation of heavy metals in plants and their ability to support plant tolerance. Additionally, many transgenic plants with an elevated resistance and improved uptake of heavy metals have been developed for improving phytoremediation. Once the rate-limiting steps for the uptake, translocation, and detoxification of HMs in hyperaccumulating plants were identified as key processes, more effective constructions of transgenic plants improved the application of phytoremediation technologies.

Compared with those in non-hyperaccumulating plants, an increased accumulation of HMs in hyperaccumulating plants can be attributed to the efficient uptake of HMs into plant root cells, reduced sequestration of metals in root vacuoles, rapid and efficient loading of xylem to transport HMs into the stem, and improved ability to detoxify and sequester HMs in leaf vacuoles. Additionally, a number of hyperaccumulating plants with rhizosphere bacteria can increase heavy metal uptake from the soil by bioactivating the rhizosphere, which might be enriched by the release of organic and inorganic compounds such as organic acids, sugars, phytochelatins, amino acids, and nitrogen compounds [90,91]. For example, some rhizosphere bacteria, such as Pseudomonas and Geobacter, can reduce oxidized Mn^4+^ to Mn^2+^, making it more available to plants, since the Mn^2+^ form is the most soluble in soil [48]. In contrast to non-hyperaccumulator plants, which mainly store heavy metals in their roots, hyperaccumulators transport and store these metals in their aboveground parts. This movement of heavy metals occurs through the xylem, from the roots to the shoots [92]. The availability of heavy metals for loading the xylem is facilitated through their low sequestration and easy exit from the vacuoles located in the root cells [35].

Notably, the main goal of the sequestration of HMs is to limit their accumulation in the cytoplasm, where they exhibit a noticeable toxic effect, and to move them to compartments where they have little activity, mainly the cell wall and vacuoles [93]. Vacuoles are the most preferred sites for the sequestration of heavy metals since vacuole enzymes, including phosphatases, lipases, and proteinases, are practically unaffected by HMs [94].

The accumulation of HMs occurs both in vacuoles of the epidermis and mesophyll cells [95], but the accumulation of HMs in epidermal cells is preferable since they do not contain chloroplasts; hence, there is no influence on photosynthesis.

This concept is based on the fact that the ability to hyperaccumulate is regulated by the expression of certain genes that are also present in non-hyperaccumulative plants, as described in detail [86,92]. It has been suggested that increasing the expression of certain genes can effectively increase the ability of plants to hyperaccumulate heavy metals. These genes, which exhibit elevated expression, encode metal transporters and enzymes that regulate the synthesis of metal ligands. This genetic enhancement is crucial for improving metal uptake and sequestration in hyperaccumulator plant species [86,96,97].

The efficiency of metal loading in plants is closely linked to the elevated expression of numerous genes that encode crucial proteins, enzymes, and heavy metal transport. Many of the identified changes in these elements related to the elevated expression of ZIP (Zrt1/IRT1-related proteins) transporter genes, heavy metal ATPase (HMAs) genes, and nicotianamine synthase genes were found in *A. halleri* and *N. caerulescens* model systems [98]. For example, Tahakashi et al. (2012) studied *A. thaliana* and metal hyperaccumulator plants and reported that HMAs play an important role in the translocation or detoxification of Zn and Cd in plants [87]. For example, there are nine HMA genes in rice, of which OsHMA1-OsHMA3 belong to the Zn/Co/Cd/Pb subgroup. OsHMA2 plays an important role in the root-to-shoot translocation of Zn and Cd and participates in Zn and Cd transport to developing rice seeds. OsHMA3 transports Cd and plays a role in the sequestration of Cd into vacuoles in root cells. It is suggested that modifying the expression of genes can be an effective method for reducing the Cd content in rice grains. Members of the zinc-iron permease (ZIP) family, including iron-regulated transporters (IRTs), are responsible for zinc (Zn), manganese (Mn), iron (Fe), and Cd uptake [99]. Additionally, the uptake of nickel is mainly linked to low-affinity transport systems, mainly from the ZIP family [87]. In addition, other genes and corresponding proteins play important roles in HM accumulation. Thus, metal tolerance protein 1 (TgMTP1) is suggested to play an important role in Zn hyperaccumulation in *T. goesingense* [100]. TgMTP1 likely helps to transfer Zn into the vacuole, enhancing both Zn accumulation and tolerance.

It is important to identify genes involved in the transfer and sequestration of HMs in hyperaccumulating plants and use these genes via biotechnological engineering to improve plant resistance and productivity. However, the mechanism of hyperaccumulation is complex and depends on plant species and soil parameters under local conditions; therefore, it will take much effort to implement this project.

## 6. Challenges and Limitations of Phytoremediation

HMs accumulate on the soil surface worldwide, raising significant concerns about environmental contamination. HMs such as Pb, Cd, Hg, and As not only affect human well-being but also disrupt ecosystem stability. Despite its promising potential, phytoremediation encounters certain challenges and limitations influenced by various factors, including HM availability, soil conditions, plant development, and long-term stability. The process’s efficiency depends on the nature and concentration of pollutants, the characteristics of the soil, the prevailing climatic conditions, and the choice of plant species for remediation. Additionally, phytoremediation projects typically require extended periods to achieve significant contaminant removal, which may not be feasible in situations where immediate decontamination is necessary [101].

### 6.1. Metal Availability

HMs have gained interest for their potential to endanger ecosystems and human health. Limited HM availability, particularly when HMs are present in small quantities or are bound to soil particles, significantly influences their impact on living organisms and the environment. In terms of HMs, “bioavailability” refers to the amount of HMs in the soil that plants, microorganisms, and other ecosystem organisms can absorb. Bioavailability is influenced by various factors, such as the chemical form of a metal, the metal concentration, the soil pH, and interactions with soil components [102].

HMs often occur at low concentrations in the soil and cannot be easily absorbed by plants or organisms. Although a shortage of available HM ions can impede their absorption, offering plant protection under certain conditions, it can hinder the uptake of crucial micronutrients, affecting plant growth and development. HMs strongly adhere to soil particles, making them unavailable for uptake. They may stick to the surface or become a structural component of minerals, effectively sequestering them from the soil solution. This immobilization, which is beneficial for reducing the potential of HMs to leach into groundwater or be absorbed by plants, simultaneously restricts HM mobility within the ecosystem [103].

The limited bioavailability of heavy metals (HMs) in natural ecosystems can disrupt plant nutrition and overall development, perturb microbial communities, and disturb the food chain; however, depending on the specific heavy metals, limited bioavailability can sometimes be beneficial for plants and microorganisms. In agriculture, low bioavailability results in deficiencies of vital trace elements in crops, posing challenges to food safety and crop quality. Considering human health, limited HM bioavailability can have both beneficial and detrimental effects; i.e., reducing direct exposure to toxic HMs can impede efficacy and limit the potential for phytoremediation of contaminated soils. Furthermore, innovative strategies for employing chelating agents to enhance HM bioavailability are currently being explored. Thus, it is crucial to thoroughly investigate the bioavailability of limited HMs to accurately assess the risks associated with HM contamination and devise appropriate strategies to protect ecosystems and human health [104].

### 6.2. Soil Conditions

Different plant species exhibit specific pH preferences, profoundly influencing their growth and HM uptake capabilities. For instance, certain hyperaccumulators may encounter challenges flourishing in acidic soil [105]. Consequently, understanding the pH requirements of chosen hyperaccumulators is crucial for the effective implementation of phytoremediation technology. The soil organic matter content directly influences moisture retention, nutrient availability, and microbial activity, all of which are critical for phytoremediation success. High soil organic matter content improves soil fertility, fosters hyperaccumulator development, and increases the potential of plants to absorb HMs. Organic amendments may be required to improve the quality of low organic matter soil and effectively support phytoremediation. Furthermore, it is crucial to maintain a balanced nutrient level in the soil to promote the growth and development of hyperaccumulators, as nutrient imbalances, whether excessive levels of certain elements or a deficiency of essential elements, can hamper plant growth and impair HM absorption ability [106]. Soil testing and nutrient management strategies can be used to rectify these imbalances and optimize conditions for effective phytoremediation [107].

Although certain plants exhibit resilience and may grow under diverse conditions, making them adaptable alternatives for various phytoremediation projects, the selection of hyperaccumulators should coincide with targeted site contamination and soil characteristics. When soil conditions are suboptimal for phytoremediation, soil amendment approaches such as composting, liming, and fertilization can adjust the soil suitability for hyperaccumulator growth and improve HM absorption [108]. Phytoremediation is an ongoing process necessitating continuous monitoring and adaptation to evolving soil conditions that allows the enhancement of restoration efficacy. In summary, implementing appropriate soil management strategies enables the full leveraging of phytoremediation potential to combat soil contamination and support sustainable environmental recovery.

### 6.3. Plant Growth and Biomass Productivity

In the context of current environmental challenges, the need for sustainable and ecologically benign solutions has become unprecedented. As a promising solution, phytoremediation has the main drawback of requiring the establishment and development of vigorous and flourishing plant populations. These plant populations hold significant potential for phytoremediation due to their accelerated growth rates and enhanced nutrient absorption capabilities. Well-nourished and resilient plants efficiently absorb contaminants from environmental matrices, thereby increasing overall removal efficiency. Phytoremediation is usually applied to intricate ecosystems with elevated contamination levels. Resilient plant populations can accelerate the phytoremediation process. Extensive root systems and high biomass yields ensure the effective sequestration of contaminants, thereby leading to a more pronounced reduction in contaminant concentrations. Furthermore, plant stands contribute to ecosystem stability by preventing soil erosion, improving soil structure, and promoting microbial activity and composition. All these factors create favorable conditions for the successful implementation of phytoremediation technology and contribute to the reinstatement of ecological equilibrium in contaminated areas [109]. Several obstacles can hinder the efficiency of phytoremediation, including the following:

Deficient soil nutrition can limit plant growth and the ability to absorb contaminants; consequently, proper soil enrichment and fertilization are crucial for maintaining a healthy plant population during phytoremediation.

-Plant susceptibility to pests and diseases can diminish plant growth and vitality; therefore, comprehensive pest control strategies are essential for mitigating these threats to the phytoremediation process.-Plant selection and genetic diversity can significantly influence phytoremediation success. Thus, identifying suitable hyperaccumulators with the necessary genetic traits is critical [8].

Adverse environmental conditions can decelerate the phytoremediation process, underscoring the importance of selecting plant species suited to the local climate, temperature, and water availability [110].

### 6.4. Complex Contamination and Soil Biodiversity

Complex contamination poses a particular challenge for restoration methods due to the intricacies of interactions between contaminants of different origins. Nevertheless, phytoremediation remains promising even for this type of contamination. Among the phytoremediation methods for co-occurring contaminants, phytoextraction is effective at accumulating high concentrations of HMs; however, the presence of organic contaminants can disrupt this process, necessitating careful plant selection. In the case of complex contamination, plants utilizing rhizodegradation, specifically by releasing root exudates that stimulate microbial composition and growth, are preferable for enhancing remediation. Soil microorganisms are capable of degrading organic contaminants, such as hydrocarbons [111]. Furthermore, phytostabilization can be more effective at immobilizing contaminants in the soil than attempting to extract or decompose them, especially for complex contaminants that are challenging to extract or breakdown. Soil biodiversity encompasses microorganisms, fungi, and other organisms inhabiting the soil and plays a crucial role in phytoremediation by influencing plant health and interactions with contaminants. A wide spectrum of microbial communities provides diverse metabolic pathways for degrading organic contaminants, increasing the efficacy of phytoremediation in complex contaminated soils. Soil microbial diversity contributes to nutrient cycling, ensuring that plants have access to essential nutrients for growth and contaminant uptake. Microorganisms can enhance plant resistance to contaminants by producing detoxifying substances. Soil biodiversity can improve soil structure and porosity, facilitate root growth, and aid in contaminant uptake [112]. By leveraging unique plant capabilities and soil microbial diversity, phytoremediation represents a sustainable solution for restoring ecologically impaired lands. Innovative methods to optimize phytoremediation techniques for complexly contaminated sites while ensuring a clean and healthy environment for future generations have been extensively explored.

### 6.5. Advanced Techniques in Phytoremediation

Recent advances in genetic engineering have revealed new vistas in phytoremediation, offering innovative strategies to combat soil contamination by heavy metals (HMs). Among these, CRISPR-Cas9 genome editing stands out as a transformative tool that enables precise modifications to plant genomes for improved tolerance and accumulation of HMs. Sarma et al. (2021) elucidated the potential of genome engineering, particularly through the use of CRISPR-Cas9, to develop climate-resilient phytoremediation plants [113]. Khan et al. (2023) highlighted the importance of the integration of advanced biotechnological strategies, including CRISPR-Cas9, in improving the phytoremediation efficiency of invasive plant species [109]. Moreover, the exploration of CRISPR-Cas9-mediated mutagenesis in wheat microspores by Bhowmik et al. (2018) represents a significant leap toward enhancing the genetic makeup of plants used in phytoremediation [114]. By optimizing the delivery of CRISPR-Cas9 reagents, they achieved targeted modifications, suggesting a scalable and efficient approach for cultivating plants with superior phytoremediation capabilities. The work by Basharat et al. (2018) further explored the horizon of genome editing in phytoremediation [115]. They proposed the utility of CRISPR-Cas9 in gene regulation through CRISPR interference and activation (CRISPRi and CRISPRa), offering nuanced control over gene expression related to metal tolerance and accumulation. The integration of CRISPR-Cas9 into phytoremediation research not only enhances the efficiency of contaminant removal but also contributes to the sustainable restoration of polluted environments.

## 7. Trends and Future Prospects of Phytoremediation

Phytoremediation is acknowledged as a cost-effective and eco-friendly strategy for removing pollutants from various environmental sources. This concept, while historically rooted, is continuously evolving. Challenges outlined in previous discussions, such as site, climate, and contaminant specificity, impede widespread adoption. Nevertheless, ongoing research is progressively addressing these challenges, with notable advancements in areas such as soil amendments and genetic modifications enhancing the technique’s efficacy. The growing acceptance and interest among environmental professionals and the general public highlight the potential of phytoremediation as a natural remediation solution, supported by a wealth of positive experimental outcomes [116].

Innovations in research are transforming phytoremediation into a modern, promising, and economical approach. However, further exploration is needed, particularly concerning the ecological impact of byproducts or accumulated contaminants. Enhancing the viability of hyperaccumulator plants in severely polluted areas, such as mining sites, is a crucial research direction [3]. Moreover, identifying commercial prospects, including applications in green roofing and eco-friendly construction projects, offers new avenues for economic utilization [117].

Recent studies underscore the role of nanotechnology in augmenting heavy metal (HM) processing in plants, demonstrating the potential of nanoparticles to increase HM absorption through bioengineering techniques [118]. Similarly, the synergy between plants and microbes in microbial-assisted phytoremediation has been shown to increase HM detoxification and uptake [3]. Moreover, in the future, the frequent and uncontrolled use of nanomaterials may require the use of phytoremediation to remove the nanoparticles themselves. The future of phytoremediation is closely linked to political and regulatory support, with government and environmental bodies playing a critical role in promoting this green technology by enabling policies, financial incentives, and clear regulatory guidelines (Figure 4).

This SWOT analysis highlights the strengths, weaknesses, opportunities, and threats associated with phytoremediation (Table 2). To maximize its benefits, phytoremediation should be considered part of a comprehensive remedial strategy, taking into account site-specific factors, technological advancements, and regulatory support. Despite its limitations, phytoremediation remains a crucial tool in the arsenal of sustainable environmental remediation methods [119].

## 8. Conclusions

Economic considerations, public acceptance, and policy implications highlight phytoremediation’s role as a sustainable solution for HM contamination in soil ecosystems. This study explored various aspects, including mechanisms of HM uptake and tolerance, providing practical guidance for plant species selection and site-specific assessments. Despite the potential of phytoremediation, its limitations, such as the long-term effectiveness and potential ecological impacts of introduced hyperaccumulators, must be acknowledged. The contributions of this research include refining plant selection and improving implementation strategies. Future work should focus on genetic engineering to enhance plant capabilities, explore synergies with microbial remediation, and assess long-term sustainability and ecological integration. Phytoremediation stands at the forefront of innovative environmental restoration, promising significant advancements in cleaner and healthier ecosystems. Building on this foundation, we hypothesize that integrating advanced nanotechnology with phytoremediation could significantly improve the efficiency of heavy metal uptake. Future studies should investigate this synergy, focusing on optimizing nanomaterials for enhanced metal absorption and stress tolerance in hyperaccumulator species.

## Figures and Tables

**Figure 1 plants-13-01534-f001:**
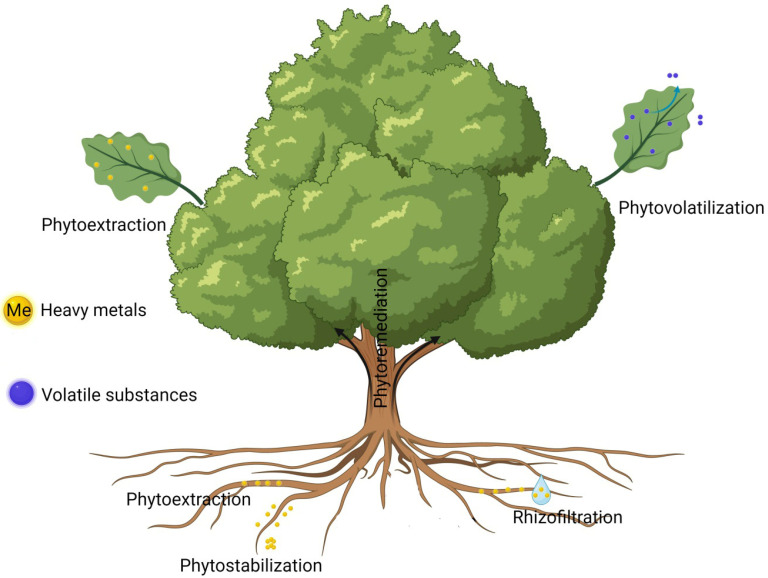
An analysis of processes and strategies for utilizing heavy metals accumulated in common plant species. Phytoextraction involves plants extracting and translocating pollutants from soil to above-ground parts and roots. Phytovolatilization refers to the use of plants to absorb heavy metal pollutants and transform them into volatile, less hazardous chemical species via transpiration. Some of the heavy metals, such as Hg and Se, may exist in the environment as gaseous species. Rhizofiltration is the method based on plant roots’ capacity to absorb and sequester metal pollutants from the water. Phytostabilization binds soil contaminants to plant roots.

**Figure 2 plants-13-01534-f002:**
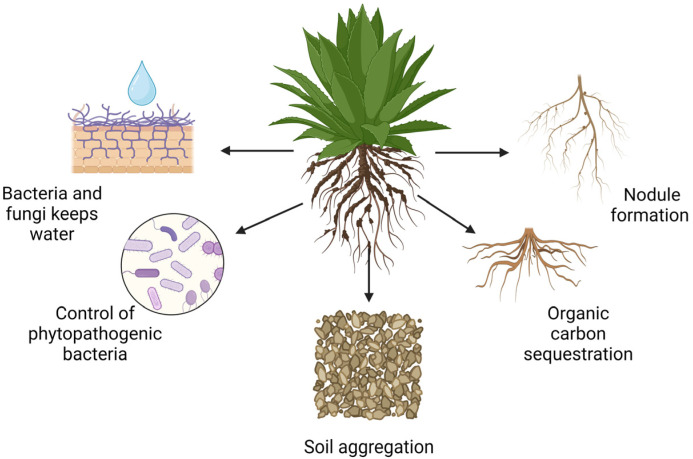
The critical role of hyperaccumulator plants in desert and semidesert ecosystems. The synergistic interaction between bacterial and fungal hyphae with soil particulates, crucial for maintaining soil moisture and structure. This enhances water infiltration and root penetration. It also illustrates the symbiotic relationship between flora and nitrogen-fixing bacteria, which form nodules on roots and convert atmospheric nitrogen into bioavailable compounds. Additionally, the figure shows organic carbon sequestration through photosynthesis and root development, contributing to soil organic matter formation [81].

**Figure 3 plants-13-01534-f003:**
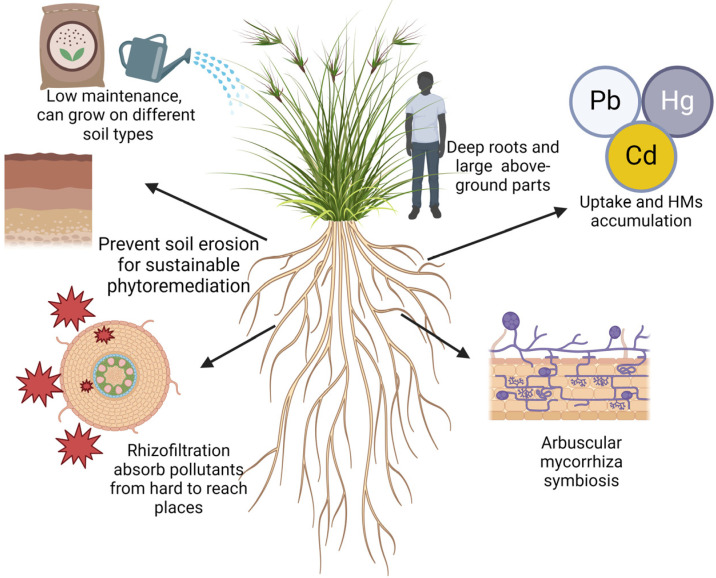
Ecological benefits and phytoremediation potential of *Chrysopogon zizanioides*. *C. zizanioides*, adaptable to various soil types and low maintenance, supports ecological restoration. Its extensive root system combats soil erosion, preserves soil structure, and enhances nutrient-rich topsoil. Through rhizofiltration, the roots absorb pollutants, purifying soil and water. The plant’s symbiotic relationship with arbuscular mycorrhizal fungi boosts nutrient uptake and aids in heavy metal stabilization. It effectively accumulates heavy metals such as lead (Pb), mercury (Hg), and cadmium (Cd), demonstrating its potential for detoxifying contaminated sites [84].

**Figure 4 plants-13-01534-f004:**
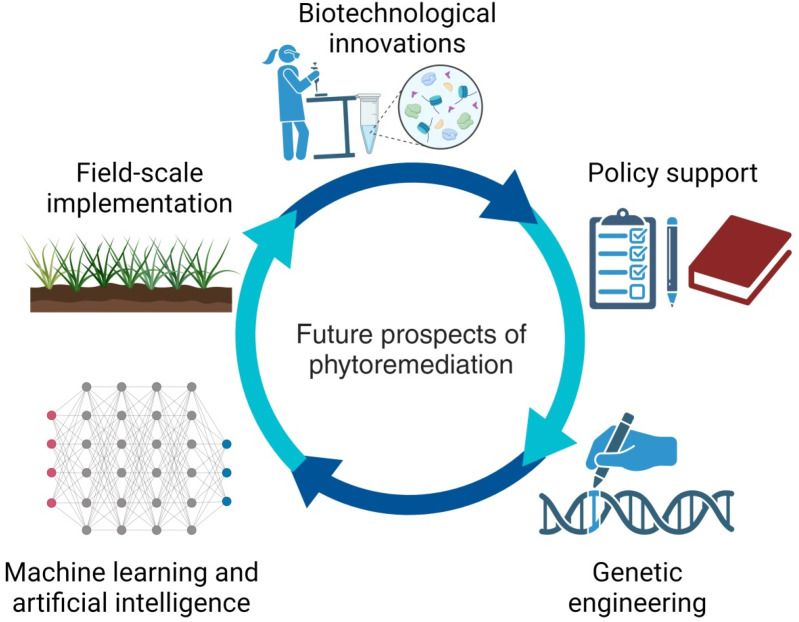
The future of phytoremediation hinges on integrating biotechnological innovations and policy support. Genetic engineering develops hyperaccumulator species and tailors pollutant metabolism. Policy development ensures regulatory frameworks for application. Machine learning and AI optimize strategies and predict efficacy, with field-scale implementation testing these methods. This approach combines technological advancements, practical applications, policy, and genetic modifications, driving continuous improvement in phytoremediation for sustainable environmental cleanup.

**Table 1 plants-13-01534-t001:** Advantages and limitations of phytoremediation methods.

Methods	Advantages	Limitations	Mechanism	Contaminants	Refs.
Phytoextraction	Cost-effective and eco-friendly. Suitable for both large- and small-scale remediation projects. A sustainable way to reduce HM concentrations in soil.	Limited to sites contaminated with specific HMs. Requires careful selection of appropriate accumulator plants. May take several decades to achieve desired results.	Hyperaccumulation in harvestable plant tissues	Elements: Pb, Zn, Au, Co, Cr, Ni, Hg, Mo, Ag, and Cd Radionuclides: Pb, Sr, U, and Cs	[12]
Rhizofiltration	Efficient for remediation of contaminated water. Suitable for various water bodies, including ponds, rivers, and constructed wetlands. Lower operating costs.	Limited to water bodies with suitable vegetation. May require careful monitoring to prevent plant overgrowth. Efficiency depends on water flow rates and environmental factors.	Rhizosphere accumulation through precipitation, sorption	Inorganic: Cr, Cd, Cu, and Ni	[13]
Phytostabilization	Reduced risk of contaminant migration. Can be combined with other phytoremediation techniques. Lower maintenance requirements.	Contaminant specific. Efficiency depends on plant selection and soil conditions. May take a longer time to achieve desired results.	Sorption, precipitation, and chelation	Inorganic: Cu, As, Cr, Zn, Cd, and Pb	[15]
Phytovolatilization	Effective for airborne contaminants. Suitable for in/outdoor remediation projects. Aids to reduce the overall contamination level of the environment.	Limited to contaminants that can be transformed into gas form. May require careful selection of suitable plants. The release of gases into the atmosphere may raise air quality concerns.	Pollutant eradication	Organic: phenols, munitions herbicides, chlorinated solvents	[16]

**Table 2 plants-13-01534-t002:** SWOT analysis of phytoremediation [120].

STRENGTHS (S)	OPPORTUNITIES (O)
- Ecologically safe - Cost-effectiveness - improve the visual appeal of contaminated sites by introducing vegetation. - Provide sustainable solutions by gradually improving the quality of soil and groundwater.	- Ongoing research led to the discovery of plant species and biotechnological techniques, further enhancing phytoremediation’s effectiveness. - Phytoremediation can be synergistically combined with other recovery methods, such as soil correction or microbial treatment. - Raising awareness and evolving regulations may create more opportunities to adopt phytoremediation.
WEAKNESSES (W)	THREATS (T)
- Phytoremediation may not be effective for highly toxic substances or in extremely dry or cold climates. - The slow pace of phytoremediation may not be suitable for sites requiring rapid cleanup. - Difficulties in selecting suitable plant species for phytoremediation can hinder the process, as some plants may struggle to thrive in contaminated soil. - There is a risk of contaminants being absorbed and accumulating at high trophic levels, potentially causing harm to animals or humans.	- Developing regulations and permitting procedures can cause difficulties in the phytoremediation implementation. - Stakeholders can impede the widespread phytoremediation adoption as a preferred method for remediation. - The success of phytoremediation can vary based on the specific site conditions. - Some phytoremediation projects may demand significant resources in terms of time, labor, and ongoing maintenance. - Phytoremediation contends with competition from established and emerging remediation technologies, potentially limiting its market share.

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
