# Peer review of "Reducing Heavy Metal Contamination in Soil and Water Using Phytoremediation"

_plants, 2024, doi:10.3390/plants13111534_

Round 1
Reviewer 1 Report
Comments and Suggestions for Authors
The review focuses on various aspects concerning phytoremediation as a sustainable and cost-effective approach for the detoxification of contaminated soils using plants. This is an important and timely topic which fits the scope of the journal. The review is well-structured. It covers different types of phytoremediation technologies, metal uptake by plants and plant tolerance mechanisms, plant selection for phytoremediation, mechanisms of hyperaccumulation, challenges and limitations in phytoremediation as well as trends and future prospects in the field. The tables are informative. The citations are relevant.
There are some minor remarks:
Rhizofiltration is usually considered as one of the types of phytofiltration. Phytofiltration includes rhizofiltration and blastofiltration.
One of the important examples of the use of phytovolatilization is applying transgenic plants for soil decontamination from organic mercury-containing compounds (methylmercury, dimethylmercury and methylmercury chloride). It may be worth including this in the review. See e.g. Hussein et al., 2007 etc.
Line 68: “high concentrations can be” – the phrase is not finished.
Lines 150-152: “This section may be divided by subheadings…” Please delete.
Lines 248-249: “Phytovolatilization is a specialized phytoremediation method aimed at reducing volatile contaminants, including organic compounds and HMs vapor, by converting them into gaseous forms” This is not clear. Please consider revising. Phytovolatilization involves the uptake of contaminants by plant roots and their conversion to a gaseous state, and release into the atmosphere.
Lines 293-294: “root exudation, which comprises organic acids, sugars, and amino acids” In fact, root exudates contain a larger variety of substances, which can not only increase but also decrease the availability of metals to plants.
Line 326: “across cellular compartments, including the cytoplasm and xylem” – this is unclear, please revise
Lines 328-329: “in plants such as A. thaliana and several other hyperaccumulator species” – A. thaliana is not a hyperaccumulator species
Line 350-351: Which NRAMP transporters are meant here? Please specify, since some NRAMP transporters are located at the plasma membrane (e.g. NRAMP1).
The phytochelatins are described first in the section ‘Intracellular ligands of heavy metals’ and then in a separate section, ‘Phytochelatins’. Maybe, it would be better to make it a subsection then. Or just merge the two sections.
Lines 459-460: “which have a generic structure called n-Gly” - Please consider revising, this is not entirely correct. It should be (γ-Glu-Cys)n-Gly.
Line 489-490; “Enzymatic detoxification is the foremost among these mechanisms, wherein plants employ a range of enzymes as frontline defenses against invading HMs.” - Providing some examples would strengthen this point.
Lines 491-492: “Metallothioneins and glutathione-S-transferases play pivotal roles in binding and sequestering HMs” - Please consider revising
Lines 510-512 and lines 516-519 contain one and the same idea.
Line 531: Thlaspi caerulescens has been moved to the genus Noccaea and is now Noccaea caerulescens
Line 569-572: “Nevertheless, certain exotic plant species, called metallophytes, contribute to soil regeneration by absorbing specific HMs. Metallophytes can be classified as indicator, exclusive, or hyperaccumulator species. Indicators concentrate HMs in AGB, while exclusive species limit HM accumulation in different tissues.” Please consider revising. Please explain which plant species are called metallophytes, indicators, excluders and hyperaccumulators. There can be excluder metallophyte species, e.g. metallicolous populations of Silene vulgaris, but most of excluder species are not metallophytes.
Lines 588-590: “The phytoremediation potential of halophytes is valuable because of their metalloids and their economic efficiency [67]. Heavy metals are considered essential agricultural tools for improving contaminated soils because they can accumulate significant amounts of HMs” – Please consider revising.
Adding some examples of hyperaccumulator halophytic and xerophytic plant species seems relevant (Section ‘Plant selection’).
Lines 598-600: “The recognition of hyperaccumulator halophytic or xerophytic plants as primary indicators for the protection of soil-ecological functions in combating global desertification and soil contamination” – This is unclear, please consider revising.
Figure 2 – misprint: ‘photopathogenic’ should be ‘phytopathogenic’; It is a bit unclear how hyperaccumulator plants help to ‘control phytopathogenic bacteria’. How widespread are hyperaccumulator legume plants capable of nodule formation in desert and semidesert ecosystems? Please provide some examples.
Lines 629-630: Shouldn`t it be “intercropping” instead of “cropping”?
Figure 3 “Uptake an HMs accumulation” – Please correct.
Lines 661-667: This sentence has been repeated twice.
Lines 670-671: “Arabidopsis, Halleri” should be Arabidopsis halleri
Lines 676-677: “the hyperaccumulation of heavy metals in the soil of hyperaccumulator plants” – Please rephrase.
Lines 702-704: “…hyperaccumulators store the absorbed HMs in their above- ground parts when the flow of HMs moves via the xylem from the shoot” – Please consider rephrasing.
Line 711: Please add a reference.
Line 719: “These genes exhibit elevated expression of several metal homeostasis genes” - Please consider rephrasing.
Line 723: “genes encoding key xylem and HM transport proteins and enzymes” – Please consider rephrasing.
Comments on the Quality of English LanguageMinor editing of English language is required.
Author Response
- Rhizofiltration is usually considered one type of phytofiltration. Phytofiltration includes rhizofiltration and blastofiltration.
Answer: We added the following paragraph:
Phytofiltration encompasses techniques such as rhizofiltration and blastofiltration, aimed at removing contaminants from water using plant systems. Rhizofiltration specifically involves the use of plant roots to absorb metals and other pollutants directly from water. On the other hand, blastofiltration uses certain plant plants for the removal of heavy metals such as lead and cadmium from water, leveraging the plants’ ability to absorb and decrease the concentration of these metals, thus purifying the water effectively. Both techniques offer ecological and efficient methods for treating contaminated water bodies (Yan, A., Wang, Y., Tan, S. N., Mohd Yusof, M. L., Ghosh, S., & Chen, Z. (2020). Phytoremediation: a promising approach for revegetation of heavy metal-polluted land. Frontiers in plant science, 11, 359. ).
- One of the important examples of phytovolatilization is the application of transgenic plants for soil decontamination from organic mercury-containing compounds (methylmercury, dimethylmercury and methylmercury chloride). It may be worth including this in the review. See, e.g., Hussein et al., 2007.
Answer: We added the following paragraph:
One of the important examples of phytovolatilization is the application of transgenic plants for soil decontamination from organic mercury-containing compounds (methylmercury, dimethylmercury and methylmercury chloride) Hussein, H. S., Ruiz, O. N., Terry, N., & Daniell, H. (2007). Phytoremediation of mercury and organo-mercurials in chloroplast transgenic plants: enhanced root uptake, translocation to shoots, and volatilization. Environmental science & technology, 41(24), 8439-8446.
- Line 68: “high concentrations can be” – the phrase is not finished.
Answer: We improved the sentences.
HMs are classified into essential and nonessential HMs based on their roles in the natural world. While essential HMs are vital in small quantities, high concentrations can be toxic or detrimental to human health.
- Lines 150-152: “This section may be divided by subheadings…” Please delete.
Answer: This has been done.
- Lines 248-249: “Phytovolatilization is a specialized phytoremediation method aimed at reducing volatile contaminants, including organic compounds and HMs vapour, by converting them into gaseous forms for safe release into the atmosphere” This is not clear. Please consider revising. Phytovolatilization involves the uptake of contaminants by plant roots and their conversion to a gaseous state and release into the atmosphere.
Answer: We improved the paragraph.
Phytovolatilization is a phytoremediation process in which plants absorb contaminants through their roots and convert them into a gaseous form. These contaminants are then released into the atmosphere through plant transpiration. This method is particularly useful for dealing with certain organic compounds and volatile heavy metals.
- Lines 293-294: “A pivotal facet of rhizosphere dynamics is root exudation, which comprises organic acids, sugars, and amino acids and is crucial for orchestrating soil modifications” In fact, root exudates contain a larger variety of substances, which can not only increase but also decrease the availability of metals to plants.
Answer. A pivotal facet of rhizosphere dynamics is root exudation, which includes a diverse array of substances, such as organic acids, sugars, amino acids, phenols, and enzymes. These exudates are crucial for modifying the soil environment, potentially increasing or decreasing the availability of metals to plants, thus influencing nutrient uptake and contaminant mobility.
- Line 326: “across cellular compartments, including the cytoplasm and xylem” – this is unclear, please revise
Answer: We improved the paragraph.
Heavy metal transporters (HMTs) are specialized proteins that play a critical role in the movement of metal ions within plant cells, facilitating their transfer through the xylem and among various cellular compartments.
- Lines 328-329: “in plants such as A. thaliana and several other hyperaccumulator species” – A. thaliana is not a hyperaccumulator species
Answer: We improved the paragraph.
Although A. thaliana is not a hyperaccumulator, other plants within the Brassicaceae family, such as Noccaea caerulescens (formerly known as Thlaspi caerulescens) and Alyssum species, are known for their hyperaccumulating properties. These plants can absorb and accumulate extremely high levels of heavy metals such as zinc, nickel, and cadmium, making them valuable for phytoremediation efforts.
- Line 350-351: Which NRAMP transporters are meant here? Please specify, since some NRAMP transporters are located at the plasma membrane (e.g. NRAMP1).
Answer: This has been done.
- The phytochelatins are described first in the section ‘Intracellular ligands of heavy metals’ and then in a separate section, ‘Phytochelatins’. Maybe, it would be better to make it a subsection then. Or just merge the two sections.
Answer: We agree with the reviewer; however, the renouncement of these paragraphs will lead to the formation of a large division that is difficult for the reader.
- Lines 459-460: “which have a generic structure called n-Gly” - Please consider revising, this is not entirely correct. It should be (γ-Glu-Cys)n-Gly.
Answer: We agree. It is done.
- Line 489-490; “Enzymatic detoxification is the foremost among these mechanisms, wherein plants employ a range of enzymes as frontline defenses against invading HMs.” - Providing some examples would strengthen this point.
Answer: We agree. It is done.
Examples of these enzymes include glutathione S-transferases, which help in conjugating toxic metals to glutathione, facilitating their sequestration; superoxide dismutases, which mitigate oxidative stress caused by metals; and phytochelatin synthases, which synthesize phytochelatins that bind metals and enhance their vacuolar storage. (Jozefczak, M., Remans, T., Vangronsveld, J., & Cuypers, A. (2012). Glutathione is a key player in metal-induced oxidative stress defenses. International journal of molecular sciences, 13(3), 3145-3175.)
- Lines 491-492: “Metallothioneins and glutathione-S-transferases play pivotal roles in binding and sequestering HMs” - Please consider revising
Answer: This has been done.
Metallothioneins and glutathione-S-transferases are crucial for the detoxification of heavy metals within plants. Metallothioneins bind to heavy metals, reducing their reactivity and toxicity. Glutathione-S-transferases, on the other hand, facilitate the conjugation of toxic metals with glutathione, aiding in their sequestration and removal from sensitive cellular areas.
- Lines 510-512 and lines 516-519 contain one and the same idea.
Answer: We ruled out duplication.
- Line 531: Thlaspi caerulescenshas been moved to the genus Noccaea and is now Noccaea caerulescens
Answer: This has been done.
- Lines 569-572: “Nevertheless, certain exotic plant species, called metallophytes, contribute to soil regeneration by absorbing specific HMs. Metallophytes can be classified as indicator, exclusive, or hyperaccumulator species. Indicators concentrate HMs in AGB, while exclusive species limit HM accumulation in different tissues.” Please consider revising. Please explain which plant species are called metallophytes, indicators, excluders and hyperaccumulators. There can be excluder metallophyte species, e.g., metallicolous populations of Silene vulgaris, but most excluder species are not metallophytes.
Answer: We improved the paragraph.
Metallophytes are plants that are well adapted to HM soils and are able to survive in heavy metal-rich soils. Metallophytes can be divided into several categories: (a) metal excluders, which accumulate HMs mainly in roots; (b) metal indicators, which accumulate HMs in their aerial parts; and (c) metal accumulators, which accumulate high HM concentrations mainly in aboveground plant parts, such as shoots and leaves (Zaghloul, M. (2020). Phytoremediation of heavy metals principles, mechanisms, enhancements with several efficiency enhancer methods and perspectives: A Review. Middle East J, 9(1), 186-214.). Examples include Alyssum bertolonii, which hyperaccumulates nickel, and Thlaspi caerulescens (now Noccaea caerulescens), which is known for absorbing high levels of zinc and cadmium. Armeria maritima also thrives in environments rich in copper and lead. While excluders such as certain populations of Silene vulgaris limit metal uptake to avoid toxicity, they often grow in metal-rich soils but are not necessarily metallophytes. These plants are valuable for their potential in phytoremediation, the process of using plants to clean up soil and water contaminated with metals.
- Lines 588-590: “The phytoremediation potential of halophytes is valuable because of their metalloids and their economic efficiency [67]. Heavy metals are considered essential agricultural tools for improving contaminated soils because they can accumulate significant amounts of HMs” – Please consider revising.
Answer: We improved the paragraph.
The phytoremediation potential of halophytes is significant due to their ability to thrive in saline environments and economically manage metal contamination. These plants are effective at accumulating substantial amounts of heavy metals, offering a sustainable solution for improving soil quality in contaminated areas.
- Adding some examples of hyperaccumulator halophytic and xerophytic plant species seems relevant (Section ‘Plant selection’).
Answer: We improved the paragraph.
The incorporation of examples of hyperaccumulator halophytic and xerophytic plants can indeed enhance our understanding of phytoremediation. For instance, Salsola kali is a halophyte known for accumulating heavy metals, while Atriplex canescens, a xerophytic species, is also capable of heavy metal uptake in contaminated soils. These species not only tolerate extreme conditions but also assist in the recovery of degraded environments through the accumulation of contaminants Van Oosten, M. J., & Maggio, A. (2015). Functional biology of halophytes in the phytoremediation of heavy metal contaminated soils. Environmental and experimental botany, 111, 135-146. Ouaini, A., Yssaad, H. A., Nouri, T., Nani, A., & Benouis, S. (2023). Influence of combined stress by salinity (NaCl) and heavy metals (Pb (NO 3) 2) on proline, chlorophyll and lead accumulation in the tissues of Atriplex canescens (Pursh) Nutt. Agricultural Science & Technology (1313-8820), 15(2).
- Lines 598-600: “The recognition of hyperaccumulator halophytic or xerophytic plants as primary indicators for the protection of soil-ecological functions in combating global desertification and soil contamination” – This is unclear, please consider revising.
Answer: We improved the paragraph.
The role of hyperaccumulator plants, particularly halophytic and xerophytic species, is crucial for protecting soil ecological functions. These plants are recognized for their ability to combat global issues such as desertification and soil contamination by accumulating heavy metals, thereby contributing to ecological restoration and soil sustainability stability. This makes them valuable indicators of environmental health and sustainability.
- Figure 2 – misprint: ‘photopathogenic’ should be ‘phytopathogenic’; It is a bit unclear how hyperaccumulator plants help to ‘control phytopathogenic bacteria’. How widespread are hyperaccumulator legume plants capable of nodule formation in desert and semidesert ecosystems? Please provide some examples.
Answer: This has been done. We added this paragraph.
Hyperaccumulator plants are primarily recognized for their ability to remove heavy metals from soils, not for controlling phytopathogenic bacteria. However, certain beneficial interactions between hyperaccumulators and soil microorganisms might indirectly affect soil pathogen populations. However, hyperaccumulator legume plants capable of nodule formation in desert and semidesert ecosystems are not very common. An example is Astragals, a genus known for some species that can accumulate selenium and are found in arid regions. These plants use their nodulation capabilities to fix nitrogen while also managing to grow in metal-rich soils, contributing to soil health in challenging environments. Zahran, H. H. (1999). Rhizobium-legume symbiosis and nitrogen fixation under severe conditions and in an arid climate. Microbiology and molecular biology reviews, 63(4), 968-989. Alford, É. R., Pilon‐Smits, E. A., Fakra, S. C., & Paschke, M. W. (2012). Moreover, selenium hyperaccumulation by Astragalus (Fabaceae) does not inhibit root nodule symbiosis. American Journal of Botany, 99(12), 1930-1941.
- Lines 629-630: Shouldn`t it be “intercropping” instead of “cropping”?
Answer: This has been done.
- Figure 3 “Uptake an HMs accumulation” – Please correct.
Answer: This has been done.
- Lines 661-667: This sentence has been repeated twice.
Answer: This has been done.
- Lines 670-671: “Arabidopsis, Halleri” should be Arabidopsis halleri
Answer: This has been done.
- Lines 676-677: “The key processes associated with the hyperaccumulation of heavy metals in the soil of hyperaccumulator plants include the following” – Please rephrase.
Answer: This has been done.
The key processes involved in the hyperaccumulation of heavy metals by hyperaccumulator plants in soil include the following:
- Lines 702-704: “…hyperaccumulators store the absorbed HMs in their above- ground parts when the flow of HMs moves via the xylem from the shoot” – Please consider rephrasing.
Answer: This has been done.
In contrast to nonhyperaccumulator plants, which mainly store heavy metals in their roots, hyperaccumulators transport and store these metals in their aboveground parts. This movement of heavy metals occurs through the xylem, from the roots to the shoots.
- Line 711: Please add a reference.
Answer: This has been done.
Broadley, M. R., White, P. J., Hammond, J. P., Zelko, I., & lux, A. (2007). Zinc in plants. New phytologist, 173(4), 677-702.
- Line 719: “These genes exhibit elevated expression of several metal homeostasis genes” - Please consider rephrasing.
Answer: This has been done.
It has been suggested that increasing the expression of certain genes can effectively increase the ability of plants to hyperaccumulate heavy metals. These genes, which exhibit elevated expression, encode metal transporters and enzymes that regulate the synthesis of metal ligands. This genetic enhancement is crucial for improving metal uptake and sequestration in hyperaccumulator plant species.
- Line 723: “genes encoding key xylem and HM transport proteins and enzymes” – Please consider rephrasing.
Answer: This has been done.
The efficiency of metal loading in plants is closely linked to the elevated expression of numerous genes that encode crucial proteins, enzymes and heavy metal transporters
- Minor editing of English language is needed.
Answer: This has been done.
Reviewer 2 Report
Comments and Suggestions for Authors
Dear Authors/Editors,
the review paper about the purification of heavy metal-contaminated soil using plants is a very good presentation of the current issues and knowledge in this field. In my opinion, the manuscript is even a bit too extensive, but this does not negatively affect its overall appearance. The information presented in the work is an important study both from the point of view of ecological safety and development of technologies for environmental remediation. Only minor suggestions are included in the attached PDF file.
Kind regards
Reviewer

Author Response
We appreciate the reviewer's positive evaluation of our manuscript and his insightful comments. We have accepted all the feedback and made the necessary revisions as suggested by the reviewer.
Reviewer 3 Report
Comments and Suggestions for Authors
Please check the attachment

None
Author Response
- Figure captions – The captions of all figures are very extensive. Try to put more concise captions with the necessary details of what is present, but avoid the description of processes that are more suitable for the main text;
Answer: This has been done. We improve the figures cations.
- References – In my opinion, as a review paper are missing references in some points of the text, which I consider relevant to introduce (e.g.: Lines 181-189; 235-242; 334- 339; 391-395; 791-792; 799-802; 808-810; 833-841; 860-867; 912-914).
Answer: This has been done. We have added the latest links to all of these paragraphs.
Montreemuk, J., Stewart, T. N., & Prapagdee, B. (2023). Bacterial-assisted phytoremediation of heavy metals: Concepts, current knowledge, and future directions. Environmental Technology & Innovation, 103488
Kafle, A., Timilsina, A., Gautam, A., Adhikari, K., Bhattarai, A., & Aryal, N. (2022). Phytoremediation: Mechanisms, plant selection and enhancement by natural and synthetic agents. Environmental Advances, 8, 100203.
Yan, A., Wang, Y., Tan, S. N., Mohd Yusof, M. L., Ghosh, S., & Chen, Z. (2020). Phytoremediation: a promising approach for revegetation of heavy metal-polluted land. Frontiers in plant science, 11, 359
De Caroli, M., Furini, A., DalCorso, G., Rojas, M., & Di Sansebastiano, G. P. (2020). Endomembrane reorganization induced by heavy metals. Plants, 9(4), 482.).
Goncharuk, E. A., & Zagoskina, N. V. (2023). Heavy metals, their phytotoxicity, and the role of phenolic antioxidants in plant stress responses with focus on cadmium. Molecules, 28(9), 3921)
Zhang, L., Liu, Z., Song, Y., Sui, J., & Hua, X. (2024). Advances in the Involvement of Metals and Metalloids in Plant Defense Response to External Stress. Plants, 13(2), 313
Skuza, L., Szućko-Kociuba, I., Filip, E., & Bożek, I. (2022). Natural molecular mechanisms of plant hyperaccumulation and hypertolerance towards heavy metals. International Journal of Molecular Sciences, 23(16), 9335.
Rangel, T. S., Santana, N. A., Jacques, R. J. S., Ramos, R. F., Scheid, D. L., Koppe, E., ... & de Oliveira Silveira, A. (2023). Organic fertilization and mycorrhization increase copper phytoremediation by Canavalia ensiformis in a sandy soil. Environmental Science and Pollution Research, 30(26), 68271-68289.
Liu, Y. Q., Chen, Y., Li, Y. Y., Ding, C. Y., Li, B. L., Han, H., & Chen, Z. J. (2024). Plant growth-promoting bacteria improve the Cd phytoremediation efficiency of soils contaminated with PE–Cd complex pollution by influencing the rhizosphere microbiome of sorghum. Journal of Hazardous Materials, 469, 134085.
Zhou, P., Adeel, M., Shakoor, N., Guo, M., Hao, Y., Azeem, I., ... & Rui, Y. (2020). Application of nanoparticles alleviates heavy metals stress and promotes plant growth: An overview. Nanomaterials, 11(1), 26).
- Title – Consider that phytoremediation is just a technology that can reduce the impacts of contaminants, but that has several limitations, the use of the term “Purifications” seems me a bit disproportionate;
Answer: This has been done. We improve the title «Reducing Heavy Metal Contamination in Soil Using Phytore-mediation»
- Lines 64–66 – I think is missing “ore exploitation” in the list of anthropogenic activities present;
Answer: This has been done.
- Lines 68–69 – “high concentrations can be” the sentence is unfinished;
Answer: This has been done. «HMs are classified into essential and nonessential HMs based on their roles in the natural world. While essential HMs are vital in small quantities, high concentrations can be toxic or detrimental to human health.»
- Lines 102–104 – If we consider organic contaminants, is missing phytodegradation, as you also show in figure 1;
Answer: This has been done.
- Section 2 – Is missing also the description of phytostimulation;
Answer: This has been done.
And phytostimulation leverages the symbiotic relationship between plants and soil microorganisms, which can improve the bioavailability of heavy metals and stimulate plant growth. Techniques include the use of specific plant species, genetic engineering, and the application of growth-promoting rhizobacteria to increase the efficiency of phytoremediation processes (Montreemuk, J., Stewart, T. N., & Prapagdee, B. (2023). Bacterial-assisted phytoremediation of heavy metals: Concepts, current knowledge, and future directions. Environmental Technology & Innovation, 103488.).
- Figure 1 (caption) – You describe this figure as phytoremediation processes for heavy metals, but in fact you also present types of phytoremediation that are used only for other type of contaminants (e.g. phytodegradation);
Answer: This has been done. We improve the figure captions.
- When you refer a reference by authors and year in the text, put parentheses in the reference year;
Answer: This has been done.
- Table 1 – In the limitations of phytoextraction it is not necessary that phytoremediation is made with hyperaccumulators, can also be just accumulators;
Answer: This has been done.
- Line 157 – As this is a review paper and you introduce the term “hyperaccumulation”, I think is important to present a definition of this;
Answer: This has been done.
Hyperaccumulation is the process by which certain plants, known as hyperaccumulators, absorb and concentrate exceptionally high levels of heavy metals from the soil into their tissues. These plants can tolerate and sequester metals such as cadmium, nickel, lead, and zinc, often to levels 100 times greater than typical plants (Montreemuk, J., Stewart, T. N., & Prapagdee, B. (2023). Bacterial-assisted phytoremediation of heavy metals: Concepts, current knowledge, and future directions. Environmental Technology & Innovation, 103488.).
- Section 2.2 – Why present the description of rhizofiltration if your review is devoted to contamination in soils and this is for remediation of waters?;
Answer: This has been done. We have corrected the title.
- Section 2.4 – Since this review is dedicated to heavy metals, the examples of phytovolatilization should be focus on these contaminants, but in fact this section only addresses organic compounds;
Answer: This has been done.
Finally, the application of phytovolatilization in eradicating organic pollutants, including chlorinated solvents according to Muthusaravanan et al. (2020), showcases the broad applicability of phytoremediation strategies [15]. One of the important examples of phytovolatilization the application of transgenic plants for soil decontamination from organic mercury-containing compounds (methylmercury, dimethylmercury and methylmercury chloride) [16].
- Line 252 – You referred “Upon absorption, plants convert contaminants into less toxic or nontoxic gases”, but this does not occur necessarily;
Answer: This has been done.
- Line 357 – Please change “and” by a comma;
Answer: This has been done.
- Lines 361, 445, 452 – Do not begin sentences by chemical symbols or abbreviation. Write it in full at the beginning of a sentence;
Answer: This has been done.
- Section 3.5 – Considering the role of phytochelating agents in detoxification they should not be addressed here? A least as a sub-chapter if you want to highlight its relevance in relation to other agents? In fact, you consider here the enzymes as the main agents in detoxification, and phytochelatins are related with these agents;
Answer: This has been done.
- Lines 539-545 – This part seems me more suitable for section 3.1. Rhizosphere interactions;
Answer: This has been done.
- Lines 588-589 – From a brief analysis of the aforementioned article, I don't think it's dedicated to halophytes;
Answer: This has been done. We have clarified this issue.
- Lines 658 and 659 – Put the (-1) in the unit “mg kg-1” above the line; • Lines 670, 671, 700 – Put in italic the name of species;
Answer: This has been done.
- Lines 673-675 – This sentence is practically the same sentence which is present in lines 577-579; avoid repetitions;
Answer: This has been done. We crossed out the repeated sentence
- Line 726 – I do not understand the symbol between the species names “A. halleri и N. caerulescens”;
Answer: This has been done.
- Lines 777-778 – The statement that is made depends on the heavy metal(s); the limited bioavailability of several heavy metals can be good for plants and microorganisms;
Answer: This has been done.
The limited bioavailability of heavy metals (HMs) in natural ecosystems can disrupt plant nutrition and overall development, perturb microbial communities, and disturb the food chain; however, depending on the specific heavy metals, limited bioavailability can sometimes be beneficial for plants and microorganisms.
- Line 846 – Change “soil diversity” by “soil biodiversity”; in fact you only focus in biodiversity
Answer: This has been done.
- Lines 1010-1011 – The title of the reference is incomplete.
Answer: This has been done.
Round 2
Reviewer 1 Report
Comments and Suggestions for Authors
The manuscript has been revised.
Comments on the Quality of English LanguageMinor editing of English language is required.
Author Response
Thanks to the reviewer for his careful attention to the manuscript. Minor English correction has been done.
Reviewer 3 Report
Comments and Suggestions for Authors
I reviewed again the paper. I looked to the author’s answers to my comments, and I check the changes in the paper. Regarding this, I leave here two notes:
1) In some parts, where the authors are supposed to change the text by a new one, this was made, but the old text version was not removed. This must be done (e.g. the old title was not removed; for example, the figure captions I do not understand if it is just to maintain the new text introduced or if it is to maintain all the text… In this case the legends are even bigger!
2) In some parts you said that accept and done the corrections suggested by me, but this is not true when looking to the manuscript. Please revise this. The parts that were not corrected I put comments in the “author’s answer” file you submitted (see attached file).

Round 3
Reviewer 3 Report
Comments and Suggestions for Authors
Now it is corrected .
Comments on the Quality of English LanguageNow it is corrected .